# Differences in Oral Lesions Associated with Tobacco Smoking, E-Cigarette Use and COVID-19 Infection among Adolescents and Young People in Nigeria

**DOI:** 10.3390/ijerph191710509

**Published:** 2022-08-24

**Authors:** Omolola Alade, Morenike Oluwatoyin Folayan, Abiola Adeniyi, Yewande Isabella Adeyemo, Afolabi Oyapero, Olubukola Olamide Olatosi, Chioma Nzomiwu, Bamidele Olubukola Popoola, Joycelyn Eigbobo, Elizabeth Oziegbe, Titus Oyedele, Maha El Tantawi, Heba Jafar Sabbagh

**Affiliations:** 1Department of Preventive and Community Dentistry, Obafemi Awolowo University, Ile-Ife 220005, Nigeria; 2Department of Child Dental Health, Obafemi Awolowo University, Ile-Ife 220005, Nigeria; 3Faculty of Dentistry, University of British Columbia, Vancouver, BC V6T 1Z4, Canada; 4Department of Child Dental Health, Bayero University, Kano 700233, Nigeria; 5Department of Preventive Dentistry, Lagos State University College of Medicine, Ikeja 100271, Nigeria; 6Department of Preventive Dental Science, Rady Faculty of Health Sciences, Dr. Gerald Niznick College of Dentistry, University of Manitoba, Winnipeg, MB R3T 2N2, Canada; 7Children’s Hospital Research Institute of Manitoba, Winnipeg, MB R3E 3P4, Canada; 8Department of Child Dental Health, Faculty of Dentistry, College of Medical Sciences, University of Calabar, Calabar 540271, Nigeria; 9Department of Child Oral Health, University of Ibadan, Ibadan 200005, Nigeria; 10Department of Child Dental Health, Faculty of Dentistry, College of Health Sciences, University of Port Harcourt, Port Harcourt 500004, Nigeria; 11Dental Unit, Department of Surgery, Benjamin Carson (Snr.) School of Medicine, Babcock University Teaching Hospital, Babcock University, Ilishan-Remo 121103, Nigeria; 12Department of Pediatric Dentistry and Dental Public Health, Faculty of Dentistry, Alexandria University, Alexandria 21527, Egypt; 13Department of Pediatric Dentistry, King Abdulaziz University, Jeddah 21589, Saudi Arabia

**Keywords:** adolescent, e-cigarette, oral lesions, medical conditions, smoking, Nigeria

## Abstract

COVID-19 infection is associated with oral lesions which may be exacerbated by tobacco smoking or e-cigarette use. This study assessed the oral lesions associated with the use of e-cigarettes, tobacco smoking, and COVID-19 among adolescents and young people in Nigeria. A national survey recruited 11–23-year-old participants from the 36 States of Nigeria and the Federal Capital Territory, Abuja. Data were collected using Survey Monkey^®^. Binary logistic regression analysis was conducted. Statistical significance was set at *p*-value less than 0.05. There were 2870 participants, of which 386 (13.4%) were tobacco smokers, 167 (5.8%) e-cigarette users, and 401 (14.0%) were both e-cigarette and tobacco users; and 344 (12.0%) had ever tested positive to COVID-19. Adolescents and young people who smoked tobacco had more than twice the odds of reporting gingival inflammation, oral ulcers, dry mouth, and changes in taste than those who did not smoke. Those who used e-cigarettes had 1.5 times higher odds of reporting oral lesions. Respondents who had COVID-19 infection had higher odds of reporting gingival inflammation and lower odds of reporting dry mouth than those who did not have COVID-19 infection. These findings were significant, and may help clinicians to screen for tobacco use and COVID-19 among adolescents and young people in Nigeria.

## 1. Introduction

Oral features of COVID-19 include bullae, de-papillated or fissured tongues, erythemae, erosions, hemorrhagic crusts, herpetiform lesions, halitosis, vesicles, pustulae, macules, papules, plaque, pigmentations, necrosis, petechiae, ulcers, spontaneous bleeding, and swellings [1]. The most common features are aphthous-like lesions, candidiasis, and oral lesions of Kawasaki-like disease [1]. The oral lesions develop in areas where angiotensin-converting enzyme 2 (ACE2) receptors bind to the severe acute respiratory syndrome coronavirus 2 (SARS-CoV-2) [2]. Oral lesions result from direct SARS-CoV-2 oral infection due to impaired immune system; or indirectly from adverse reactions to the treatments of COVID-19 [3,4,5]. Typically, lesions appear between 4 days and 12 weeks after the onset of the systemic symptoms of SARS-CoV-2 infection [1].

The oral lesions observed during the COVID-19 pandemic may also be the result of the mental health challenges associated with the pandemic. The mental health challenges associated with the pandemic may be the results of the direct and indirect impact of the public health response to the pandemic. The mental health challenges may cause an increase in tobacco smoking and e-cigarettes use among current smokers, while people who had successfully quit smoking may relapse [6,7,8]. However, concerns about of the risk of severe COVID-19 in people who smoke led to a decrease in tobacco use in low-frequency smokers [9].

Smoking-associated oral lesions include abrasions, acute necrotizing ulcerative gingivitis other periodontal conditions, black hairy tongue, burns and keratotic patches, leukoplakia, nicotinic stomatitis, palatal erosions, tooth stains, smoker’s melanosis, epithelial dysplasia, and squamous-cell carcinoma [10]. Oral lesions are linked to both tobacco smoking and e-cigarette use, although the oral symptoms associated with e-cigarettes use are less severe than those reported in tobacco smokers [11].

The pandemic also affected adolescents and young people negatively. Although the COVID-19 pandemic does not cause significant physical risk to teenagers [12,13], the fear of infection, financial constraints, and social isolation pose a threat to the mental health of many adolescents [14]. The number of adolescents using e-cigarettes and smoking cigarettes decreased during the pandemic because of several reasons, including the presence of parents in the home, poor access to cigarette products during the pandemic, and concerns about increased risk of COVID-19 [15,16]. In contrast, there are reports of some adolescents increasing smoking during the pandemic for several reasons including feelings of loneliness and isolation [9].

The oral lesions associated with COVID-19 infection may be early markers of COVID-19 infection in adolescents who are less prone to symptomatic COVID-19 infection. Smoking may also increase the risk of oral symptoms of COVID-19. However, no studies have assessed the oral health lesions associated with tobacco smoking and/or e-cigarettes use among adolescents who may have COVID-19. Being able to identify oral health indicators of tobacco smokers and e-cigarettes users who have COVID-19 will enhance the role of dentists in promoting smoking cessation [17].

The aim of this study, therefore, was to assess the oral lesions associated with the use of e-cigarettes, tobacco smoking, and COVID-19; and to identify differences in the risk indicators for self-reported oral lesions associated with the use of e-cigarettes, tobacco smoking, and COVID-19 among adolescents and young people in Nigeria. We hypothesized that there will be differences in the types of oral lesions associated with the use of e-cigarettes, tobacco smoking, and COVID-19 infection among adolescents and young people.

## 2. Materials and Methods

This was a national survey conducted between 1 November and 30 December 2021. The survey recruited study participants from the 36 States of Nigeria and the Federal Capital Territory, Abuja.

### 2.1. Ethics Approval

Ethical approval was obtained from the Institute of Public Health, Obafemi Awolowo University Health Research Ethics Committee (IPH/OAU/12/1604). The study was carried out according to the National Research Ethics Regulation and the Helsinki Declaration [18]. Written informed consent was required from the parents of participants who reported they were aged 11 to 17 years old by parental click on the informed consent form. Thereafter, written assent was required from participants who were 12–17-years-old before they could continue with the study participation. Assent was provided a click on the assent box. Respondents aged 18 to 23 years participated after giving an independent written informed consent.

The administered questionnaire began by explaining the purpose of the study, assuring participants of the confidentiality of responses, and the freedom to withdraw from the survey at any time. Study participants had to check a consent box indicating that they read the information sheet and consented to participate in the study. For minors, parents had to first check the consent box before the information sheet and assent form for the adolescent pops up for checking. All participants who indicated they were not willing to participate in the study after reading the informed consent sheet were thanked and exited from the survey.

### 2.2. Study Design, Study Participants, and Study Setting

This was a cross-sectional study that involved individuals study participants were 11–23-year-olds. Study participants filled an online survey launched using the Survey Monkey^®^ platform between the 1 November to the 30 December 2021. There were no exclusion criteria for study participation.

### 2.3. Recruitment of Study Participants

Study participants were recruited using non-probability sampling techniques: convenience sampling and respondent driven sampling. The study investigators reached out to adolescents and young adult networks, sharing their unique survey link with their contacts, and encouraging them to share it with their peers. The survey link was also posted on social media groups (Facebook, Twitter, and Instagram), email lists, and WhatsApp groups of young people. In addition, the study team recruited a diverse population of 37 adolescent and young persons (one per State and the Federal Capital Territory), trained them on the study protocol, and asked them to share their unique link with their peers. Each young person got paid 26.00 USD for recruiting 50 persons.

### 2.4. Data Collection Instrument

Data were collected using a questionnaire that was developed for a multi-country study exploring the impact of COVID-19 on smoking and oral health of adolescents and young people [19]. Survey links were created with settings that ensured it would be anonymous, participants could amend their responses before submitting, and it was not time-limited. Only one submission per electronic device was allowed. The questionnaire was developed in Arabic and translated to multiple languages including English [19]. The English version was used for data collection in Nigeria and the content validity index calculated for the finalised questionnaire was 0.87 [20]. The questions were closed-ended and took an average of 10 min to complete.

### 2.5. Study Variables

#### 2.5.1. Dependent Variable

Self-reported oral lesions: Participants identified the oral lesions they had since the COVID-19 pandemic started by reviewing a list of oral lesions and ticking the appropriate checkbox. Study participants could check multiple responses. These lesions included gum (gingival) inflammation, change in taste, oral ulcers, and dry mouth [21].

#### 2.5.2. Independent Variables

Smoking status: Participants were asked about tobacco smoking and the use of e-cigarettes using the Global Youth Tobacco Survey questionnaire [22]. Participants were asked if they were current, former, or never tobacco smokers. Respondents who ticked ‘former’ or ‘never’ were categorized as non-tobacco smokers. Respondents were also asked if they had ever used e-cigarettes (yes/no). Participants who were both tobacco smokers and used e-cigarettes were categorized as dual users.

COVID-19 status: Participants were also asked if they had ever tested positive for COVID-19 (yes/no).

#### 2.5.3. Confounders

Sociodemographic profile: Participants were asked about their ages (categorized into 11–14, 15–17, and 18–23 years old), and sex at birth (male, female, others).

Medical conditions: Data were collected about medical problems. Respondents could check-off one or more of 23 medical conditions, with an option to include other medical conditions not in the list. The medical conditions listed were pneumonia, diabetes mellitus, cancer, heart condition, hepatitis, hypertension, neurological problems, neuropathy, respiratory problems, stroke, depression, herpes, shingles, and other sexually transmitted infections, dermatologic problems, migraines, arthritis, broken bones, hearing loss, vision loss, and others. These questions were adopted from a questionnaire that had been validated for global use [23]. Participants’ responses were dichotomized into those who had no health condition (those who did not check any of the options) and those who had a health condition (anyone who ticked an option).

#### 2.5.4. Statistical Analysis

Frequencies and percentages were calculated for the study variables. The associations between the dependent and independent variables were assessed using the Chi-square test. Binary logistic regression using IBM SPSS for Windows version 22.0 (IBM Corp., Armonk, NY, USA) was used to determine the independent variables associated with oral lesions, and the effect size of the associations. Adjusted odds ratio (AORs), 95% confidence interval (CIs), and *p* values were calculated. Statistical significance was set at *p*-value less than 0.05.

## 3. Results

Table 1 shows that the study included 2870 participants, of which 1449 (50.5%) were males, 386 (13.4%) were tobacco smokers, 167 (5.8%) were e-cigarette users, 401 (14%) were both e-cigarette and tobacco users, and 344 (12.0%) had ever tested positive to COVID-19.

Adolescents and young people with a health condition (AOR: 1.394, *p* = 0.005) or COVID-19 (AOR: 1.617, *p* = 0.001) had significantly higher odds of gingival inflammation. Also, gingival inflammation was significantly associated with tobacco smoking (AOR: 4.044 *p* < 0.001), e-cigarette use (AOR: 1.507, *p* = 0.048), and dual use of tobacco and e-cigarettes (AOR: 2.620, *p* < 0.001).

Change in taste was significantly associated with tobacco smoking (AOR: 2.203, *p* < 0.001) and dual use of tobacco and e-cigarette (AOR: 3.272, *p* < 0.001) but not e-cigarette use (AOR:1.618, *p* = 0.068).

Adolescents and young people who had a medical condition (AOR: 1.799, *p* = 0.002) had significantly higher odds of oral ulcers and males (AOR: 0.589, *p* = 0.001) had significantly lower odds of oral ulcers. Oral ulcers were significantly associated with tobacco smoking (AOR: 2.568, *p* < 0.001), e-cigarette use (AOR: 1.885, *p* = 0.044), and dual use of tobacco and e-cigarettes (AOR: 2.242, *p* < 0.001).

Older adolescents and young people had significantly lower odds of dry mouth (18–23 years old: AOR: 0.618, *p* = 0.008; and 15–17 years old: AOR: 0.321, *p* < 0.001), and also those with COVID-19 had significantly lower odds of dry mouth (AOR: 0.648, *p* = 0.024). Dry mouth was significantly associated with tobacco smoking (AOR: 3.997, *p* < 0.001), use of e-cigarettes (AOR:1.964, *p* = 0.008), and dual use of tobacco and e-cigarettes (AOR: 2.881, *p* < 0.001).

## 4. Discussion

The study indicates that adolescents and young people who smoked tobacco, used e-cigarettes, or used both e-cigarettes and tobacco were more likely to report gingival inflammation, oral ulcers, and dry mouth by than those who did not smoke tobacco, use e-cigarettes, or use both e-cigarettes and tobacco during the COVID-19 pandemic. Furthermore, adolescents and young people who had ever tested positive for COVID-19 were more likely to report gingival inflammation and less likely to report dry mouth. The findings partly support the study hypothesis.

The study is the first to investigate possible oral lesions in adolescents and young persons who smoke tobacco or use e-cigarettes during the COVID-19 pandemic in Nigeria. Although the data were gathered using non-probability sampling methods, the large sample size and the national distribution of the sample are strengths. The study sample could not be recruited using a probability sampling method because of the COVID-19 epidemic [24], although the use of non-probability sampling is ideal for online surveys [25,26].

A study limitation is the self-reporting of the oral lesions, which could introduce biased reporting based on subjective rather than objective assessment. The level of the risk for bias reporting is not known as self-reports of oral lesions had not been evaluated in Nigeria though this had been done in many other countries [27,28]. Moreover, there may be recall bias as participants may not remember the oral lesions they had during the pandemic. In addition, vesicular-bullous diseases (pemphigus, pemphigoid, lichen planus) present with oral lesions that may look like the ones reported by people infected with COVID-19. The oral lesion may also result from a comorbidity with COVID-19, rather than the result of COVID per se. The altered immune response associated with COVID-19 makes this a possibility. In addition, smokers may have oral lesions because of an oral ulcer-producing disease and not because of the smoking habit. The study, however, did not include these considerations in the data collection though participants had the option of listing other medical and oral health conditions they had that was not included in the medical conditions listed. Despite these study limitations, the study highlights some important findings.

First, e-cigarette users and tobacco smokers reported gingival inflammation, oral ulcers, and dryness of the mouth. While tobacco smokers reported changes in taste, e-cigarette users did not. Reporting gingival inflammation with use of tobacco and e-cigarettes agrees with prior studies showing this association [29,30,31,32,33]. Smoking masks gingival inflammation by reducing gingival bleeding due to the vaso-constrictive effects of nicotine and lower hemorrhagic responsiveness in smokers [34,35,36,37]. It also causes increased sub-gingival temperature, reduced oxygen tension, and selective favoring of periodontopathic Gram-negative anerobes [38,39]. Moreover, tobacco compounds leak into the oral mucosa altering the form, quantity, and vascularization of the taste buds reversibly or irreversibly [40,41,42,43]. Nicotine also reduces taste sensitivity because tobacco regulates the taste signal at a central level [44]. We are unable to offer reasons for why the risk for changes in taste was higher for tobacco smoking than for e-cigarette use. If this finding is confirmed in future studies, this may indicate that e-cigarettes are potentially less harmful than tobacco smoking [45].

The oral lesions association with e-cigarette use and tobacco smoking may make adolescents and young people seek care in the dental clinic, thereby giving dental care providers the opportunity to screen and appropriate counsel for tobacco use cessation. The study findings indicate that dentists can play a significant and effective role in addressing the smoking problem among adolescents and young people though tobacco cessation counseling is challenging in young people [46]. Prior studies had, however, indicated that dentists in Nigeria are unable to counsel patients on tobacco cessation because of poor knowledge and skills [47]. Furthermore, the proportion of adolescents who utilize health services is typically low [48] and may be quite low in Nigeria where dental care is often exclusively curative [49]. A public health campaign for the cessation of e-cigarette use and tobacco smoking will require wide-scale engagement of stakeholders concerned with adolescent and young people’s health within and outside the health sector using evidence-based techniques. Future studies conducted to explore reasons for e-cigarette and tobacco use by adolescents and young people may help improve the design and implementation of tobacco cessation programs.

Second, the non-significant association between COVID-19 and taste changes is worth noting. The literature documents the link between COVID-19 and taste changes [50]. However, taste changes may not be an indicator of COVID-19 infection in adolescents and young people in Nigeria. The lack of a significant association in the present study may be because of the way the question was asked. We asked about taste changes rather than loss of taste which is what is often attributed to COVID-19. Similarly, the lack of a significant association between oral ulcers and COVID-19 is contrary to what had been earlier reported in adults [51]. This difference may also be because in the study participants’ area, young people have an immunological profile that differs from that of adults. Prior reports on COVID-19 and oral ulcers had been conducted in adults. This hypothesis needs to be studied.

Third, the lower odds of reporting dry mouth in respondents with COVID-19 compared to those without COVID-19 was unusual. Dry mouth had previously been associated with COVID-19 [52,53], possibly as a result of the neuropathic and mucotropic effects of SARS-CoV-2 on salivary gland function [3] or COVID-19-related inflammatory and infectious procedures that may induce hyposalivation [54]. The present finding is difficult to explain though it suggests that the hyposalivation or xerostomia may not be an oral feature of COVID-19 in adolescents and young people. This also needs to be explored further.

## 5. Conclusions

In conclusion, gingival inflammation, oral ulcers, change in taste, and dry mouth are oral lesions reported by adolescents and young people who smoke tobacco and use e-cigarettes. Adolescents and young people with a history of COVID-19 seem to be more likely to report gingival inflammation and less likely to report dry mouth than those without COVID-19. The differences in the current report of oral lesions associated with COVID-19 in adolescents and young people, and prior reports on oral lesions seen in adults with COVID-19 may need to be investigated further. This possible age difference in the oral features of COVID-19 may guide clinicians to screen for tobacco use and COVID-19 infection among adolescents and young people attending dental clinics in Nigeria.

## Figures and Tables

**Table 1 ijerph-19-10509-t001:** Binary logistic regression of the association between risk indicators and oral lesions during the COVID-19 pandemic among 11–23-year-olds living in Nigeria (N = 2870), * statistically significant.

Variables	Total *n* (%)	Yes *n* (%)	No *n* (%)	AOR (95% C.I.), *p* Value
**Gingival inflammation**				
**Age (years)**				
18–23	2206 (76.9)	538 (24.4)	1668 (75.6)	1.215 (0.860–1.718), 0.270
15–17	327 (11.4)	44 (13.5)	283 (86.5)	0.727 (0.459–1.152), 0.174
11–14	337 (11.7)	45 (13.4)	292 (86.6)	1.000
**Sex**				
Male	1449 (50.5)	304 (21.0)	1145 (79.0)	0.850 (0.704–1.025), 0.089
Female	1421 (49.5)	323 (22.7)	1098 (77.3)	1.000
**Medical condition**				
Yes	641 (22.3)	215 (33.5)	426 (66.5)	1.394 (1.108–1.754), 0.005 *
No	2229 (77.7)	412 (18.5)	1817 (81.5)	1.000
**COVID-19 infection**				
Yes	344 (12.0)	138 (40.1)	206 (59.9)	1.617 (1.216–2.152), 0.001 *
No	2526 (88.0)	489 (19.4)	2037 (80.6)	1.000
**Cigarette use**				
Dual use	401 (14.0)	148 (36.9)	253 (63.1)	2.620 (1.968–3.489), <0.001 *
Tobacco smoking	386 (13.4)	170 (44.0)	216 (56.0)	4.044 (3.142–5.203), <0.001 *
E-cigarette use	167 (5.8)	43 (25.7)	124 (74.3)	1.507 (1.003–2.263), 0.048 *
None	1916 (66.8)	266 (13.9)	1650 (86.1)	1.000
**Changes in taste**				
Age (years)				
18–23	2206 (76.9)	272 (12.3)	1934 (87.7)	0.966 (0.640–1.457), 0.867
15–17	327 (11.4)	24 (7.3)	303 (92.7)	0.661 (0.376–1.162), 0.151
11–14	337 (11.7)	30 (8.9)	307 (91.1)	1.000
**Sex**				
Male	1449 (50.5)	166 (11.5)	1283 (88.5)	0.983 (0.777–1.243), 0.884
Female	1421 (49.5)	160 (11.3)	1261 (88.7)	1.000
**Medical condition**				
Yes	641 (22.3)	103(16.1)	538 (83.9)	1.281 (0.966–1.700), 0.086
No	2229 (77.7)	223 (10.0)	2006 (90.0)	1.000
**COVID-19 infection**				
Yes	344 (12.0)	50 (14.5)	294 (85.5)	0.764 (0.520–1.122), 0.170
No	2526 (88.0)	276 (10.9)	2250 (89.1)	1.000
**Cigarette use**				
Dual use	401 (14.0)	89(22.2)	312 (77.8)	3.272 (2.323–4.609), <0.001 *
Tobacco smoking	386 (13.4)	64(16.6)	322 (83.4)	2.203 (1.568–3.095), <0.001 *
E-cigarettes use	167 (5.8)	21 (12.6)	146 (87.4)	1.618 (0.964–2.176), 0.068
None	1916 (66.8)	152 (7.9)	1764 (92.1)	1.000
**Oral ulcers**				
**Age (years)**				
18–23	2206 (76.9)	138 (6.3)	2068 (93.7)	0.701 (0.422–1.164), 0.170
15–17	327 (11.4)	12 (3.7)	315 (96.3)	0.487 (0.230–1.032), 0.060
11–14	337 (11.7)	20 (5.9)	317 (94.1)	1.000
**Sex**				
Male	1449 (50.5)	66 (4.6)	1383 (95.4)	0.589 (0.427–0.812), 0.001 *
Female	1421 (49.5)	104 (7.3)	1317 (92.7)	1.000
**Medical condition**				
Yes	641 (22.3)	66 (10.3)	575 (89.7)	1.799 (1.238–2.615), 0.002 *
No	2229 (77.7)	104 (4.7)	2125 (95.3)	1.000
**COVID-19 infection**				
Yes	344 (12.0)	29 (8.4)	315 (91.6)	0.823 (0.513–1.321), 0.420
No	2526 (88.0)	141 (5.6)	2385 (94.4)	1.000
**Cigarette use**				
Dual use	401 (14.0)	38 (9.5)	363 (90.5)	2.242 (1.390–3.616), 0.001 *
Tobacco smoking	386 (13.4)	40 (10.4)	346 (89.6)	2.568 (1.666–3.958), <0.001 *
E-cigarette use	167 (5.8)	14 (8.4)	153 (91.6)	1.885 (1.018–3.492), 0.044 *
None	1916 (66.8)	78 (4.1)	1838 (95.9)	1.000
**Dry mouth**				
**Age (years)**				
18–23	2206 (76.9)	308 (14.0)	1898 (86.0)	0.618 (0.434–0.881), 0.008 *
15–17	327 (11.4)	20 (6.1)	307 (93.9)	0.321 (0.184–0.561), <0.001 *
11–14	337 (11.7)	48 (14.2)	289 (85.8)	1.000
**Sex**				
Male	1449 (50.5)	194 (13.4)	1255 (86.6)	0.999 (0.799–1.248), 0.993
Female	1421 (49.5)	182 (12.8)	1239 (87.2)	1.000
**Medical condition**				
Yes	641 (22.3)	103 (16.1)	538 (83.9)	1.039 (0.788–1.370), 0.787
No	2229 (77.7)	273 (12.2)	1956 (87.8)	1.000
**COVID-19 infection**				
Yes	344 (12.0)	46 (13.4)	298 (86.6)	0.648 (0.445–0.945), 0.024 *
No	2526 (88.0)	330 (12.1)	2196 (86.9)	1.000
**Cigarette use**				
Dual use	401 (14.0)	77 (19.2)	324 (80.8)	2.881 (2.048–4.053), <0.001 *
Tobacco smoking	386 (13.4)	102 (26.4)	284 (73.6)	3.997 (2.960–5.396), <0.001 *
E-cigarette use	167 (5.8)	24 (14.4)	143 (85.6)	1.964 (1.197–3.223), 0.008 *
None	1916 (66.8)	173 (9.0)	1743 (91.0)	1.000

## Data Availability

The data that support the findings of this study are available from the corresponding author but restrictions apply to the availability of these data, which were used under license for the current study, and so are not publicly available. Data are however available from the authors upon reasonable request and with permission of Morenike Oluwatoyin Folayan.

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
