# Peer review of "Differences in Oral Lesions Associated with Tobacco Smoking, E-Cigarette Use and COVID-19 Infection among Adolescents and Young People in Nigeria"

_ijerph, 2022, doi:10.3390/ijerph191710509_

Round 1
Reviewer 1 Report
Thank you for sending us the manuscript, everything looks great for publication, your manuscript requires minor revision,and needs to add keywords after abstract .
Author Response
Thank you for sending us the manuscript, everything looks great for publication, your manuscript requires minor revision, and needs to add keywords after abstract.
Response: Thank you for your positive feedback. The minor revisions have been effected and keywords are after the abstract.
Reviewer 2 Report
O Alade and co-authors have presented the assessment of oral health lesions associated with tobacco smoking and e-cigarettes in adolescents and young people during the COVID-19 pandemic. I've enjoyed reading this well-written, thorough, important, and relevant original article. I think this manuscript is worth publishing after minor revision. Comments and queries can be found below.
- Introduction
1. Authors could add some sentences to describe % of the global population using tobacco and e-cigarettes and, and compare it to the % in Nigeria.
- Method:
1. This is a cross-sectional study and data collection from an online questionnaire. According to the regulation in your country, should you provide the approval of Institutional Review Boards (IRB)? In addition, please also check your manuscript follow the STROBE Checklist (https://www.strobe-statement.org/)
2. The questionnaire should be translated into English and uploaded as supplementary information (SI)
- Result:
1. Could the author provide more figures for binary logistic regression, not just only one Table? 2. Could the author provide more data about the severity of COVID-19? For example, how many n (%) asymptomatic (mild) cases, symptomatic cases, and server case
3. I found the typo in Table 1, i.e. the E-cigarettes use % in dry mouth section. Yes (24.4%) + No (85.6%) = 110% ??? what happens? Thus, please check all of your data are correct.
- Discussion:
1. I suggest the limitation of this study should be put in the last or penultimate paragraph
Author Response
O Alade and co-authors have presented the assessment of oral health lesions associated with tobacco smoking and e-cigarettes in adolescents and young people during the COVID-19 pandemic. I've enjoyed reading this well-written, thorough, important, and relevant original article. I think this manuscript is worth publishing after minor revision. Comments and queries can be found below.
Response: Thank you for your encouraging words
- Introduction
1. Authors could add some sentences to describe % of the global population using tobacco and e-cigarettes and, and compare it to the % in Nigeria.
RESPONSE: Thanks for the comments. We have introduced a new paragraph 3 in the introduction section to address this comment
- Method:
1. This is a cross-sectional study and data collection from an online questionnaire. According to the regulation in your country, should you provide the approval of Institutional Review Boards (IRB)?
RESPONSE: Yes, in Nigeria, you will need an ethics approval for this. We did collect ethics approval for this as indicated in the methods section
In addition, please also check your manuscript follow the STROBE Checklist (https://www.strobe-statement.org/)
RESPONSE: Thanks for highlighting this. We did follow the guidelines of the STROBE Checklist to prepare the manuscript
2. The questionnaire should be translated into English and uploaded as supplementary information (SI)
RESPONSE. The questionnaire was developed in English. We have included it as a supplementary file. English is the language of communication in Nigeria
- Result:
- Could the author provide more figures for binary logistic regression, not just only one Table?
RESPONSE: We feel all that needs to be reflected as per the data for this study are all represented in the single table. The descriptive variables are there and so are the regression analysis outcome.
2. Could the author provide more data about the severity of COVID-19? For example, how many n (%) asymptomatic (mild) cases, symptomatic cases, and server case
RESPONSE: There was no data collected on the grades of the infection. We are sorry we cannot provide this detail. We have included it as study limitation. We wrote: We also were not able to grade the severity of COVID-19 reported and thus, unable to link the oral lesions to the severity of COVID-19.
3. I found the typo in Table 1, i.e. the E-cigarettes use % in dry mouth section. Yes (24.4%) + No (85.6%) = 110% ??? what happens? Thus, please check all of your data are correct.
RESPONSE: This has been corrected to Yes (14.4%). Other errors corrected
- Discussion:
1. I suggest the limitation of this study should be put in the last or penultimate paragraph
RESPONSE: thanks for this suggestion. The STROBE checklist requires that the limitation be discussed upfront so readers can objectively assess the results. We have adhered to the STROBE guidelines.
Reviewer 3 Report
Comments and Suggestions for Authors
The manuscript entitled “Differences in oral lesions associated with tobacco smoking, e- cigarettes use and COVID-19 infection among adolescents and young people in Nigeria” presents an assessment for the oral lesions associated with the use of e-cigarettes, tobacco smoking and COVID-19 infection; and studied the differences in the risk indicators for self-reported oral lesions associated with the use of e-cigarettes, tobacco smoking and COVID-19 infection among adolescents and young people in Nigeria. The systems are interesting, and it is good that the authors apply a national survey approach.
The results are well described, and the work reports some insights that could be of interest to the scientific society. However, the paper presents some mistakes in English grammar that should be fixed.
Additional comments.
1) Page 5, table 1, the column entitled (No) can be deleted
Since it gives the complementary numbers and percentage for the previous column
I recommend that this paper be published after English editing.
Author Response
The manuscript entitled “Differences in oral lesions associated with tobacco smoking, e- cigarettes use and COVID-19 infection among adolescents and young people in Nigeria” presents an assessment for the oral lesions associated with the use of e-cigarettes, tobacco smoking and COVID-19 infection; and studied the differences in the risk indicators for self-reported oral lesions associated with the use of e-cigarettes, tobacco smoking and COVID-19 infection among adolescents and young people in Nigeria. The systems are interesting, and it is good that the authors apply a national survey approach.
RESPONSE: Thank you for your positive feedback
The results are well described, and the work reports some insights that could be of interest to the scientific society. However, the paper presents some mistakes in English grammar that should be fixed.
RESPONSE: We have carefully read the manuscript and addressed grammatical errors. We hope this version of the manuscript will be more amenable to the reviewer.
Additional comments.
1) Page 5, table 1, the column entitled (No) can be deleted
Since it gives the complementary numbers and percentage for the previous column
RESPONSE: Thanks for this suggestion. We will like to retain this column as it also serves as a reference for interpretation of the results with ease
I recommend that this paper be published after English editing.
RESPONSE: The manuscript has been edited extensively